# TLD1433 Photosensitizer Inhibits Conjunctival Melanoma Cells in Zebrafish Ectopic and Orthotopic Tumour Models

**DOI:** 10.3390/cancers12030587

**Published:** 2020-03-04

**Authors:** Quanchi Chen, Vadde Ramu, Yasmin Aydar, Arwin Groenewoud, Xue-Quan Zhou, Martine J. Jager, Houston Cole, Colin G. Cameron, Sherri A. McFarland, Sylvestre Bonnet, B. Ewa Snaar-Jagalska

**Affiliations:** 1Institute of Biology, Leiden University, 2333 CC Leiden, The Netherlands; q.chen@biology.leidenuniv.nl (Q.C.); yasmin.aydar@gmail.com (Y.A.); a.groenewoud@biology.leidenuniv.nl (A.G.); 2Leiden Institute of Chemistry, Leiden University, P.O. Box 9502, 2300 RA Leiden, The Netherlands; r.vadde@lic.leidenuniv.nl (V.R.); x.zhou@lic.leidenuniv.nl (X.-Q.Z.); 3Department of Ophthalmology, Leiden University Medical Center, 2333 ZA Leiden, The Netherlands; m.j.jager@lumc.nl; 4Department of Chemistry and Biochemistry, The University of Texas at Arlington, Arlington, TX 76019, USA; houston.cole@gmail.com (H.C.); cgcamero@uncg.edu (C.G.C.)

**Keywords:** photodynamic therapy, conjunctival melanoma, in vitro and in vivo models, application in cancer, zebrafish tumour model

## Abstract

The ruthenium-based photosensitizer (PS) TLD1433 has completed a phase I clinical trial for photodynamic therapy (PDT) treatment of bladder cancer. Here, we investigated a possible repurposing of this drug for treatment of conjunctival melanoma (CM). CM is a rare but often deadly ocular cancer. The efficacy of TLD1433 was tested on several cell lines from CM (CRMM1, CRMM2 and CM2005), uveal melanoma (OMM1, OMM2.5, MEL270), epidermoid carcinoma (A431) and cutaneous melanoma (A375). Using 15 min green light irradiation (21 mW/cm^2^, 19 J.cm^−2^, 520 nm), the highest phototherapeutic index (PI) was reached in CM cells, with cell death occurring via apoptosis and necrosis. The therapeutic potential of TLD1433 was hence further validated in zebrafish ectopic and newly-developed orthotopic CM models. Fluorescent CRMM1 and CRMM2 cells were injected into the circulation of zebrafish (ectopic model) or behind the eye (orthotopic model) and 24 h later, the engrafted embryos were treated with the maximally-tolerated dose of TLD1433. The drug was administrated in three ways, either by (i) incubating the fish in drug-containing water (WA), or (ii) injecting the drug intravenously into the fish (IV), or (iii) injecting the drug retro-orbitally (RO) into the fish. Optimally, four consecutive PDT treatments were performed on engrafted embryos using 60 min drug-to-light intervals and 90 min green light irradiation (21 mW/cm^2^, 114 J.cm^−2^, 520 nm). This PDT protocol was not toxic to the fish. In the ectopic tumour model, both systemic administration by IV injection and RO injection of TLD1433 significantly inhibited growth of engrafted CRMM1 and CRMM2 cells. However, in the orthotopic model, tumour growth was only attenuated by localized RO injection of TLD1433. These data unequivocally prove that the zebrafish provides a fast vertebrate cancer model that can be used to test the administration regimen, host toxicity and anti-cancer efficacy of PDT drugs against CM. Based on our results, we suggest repurposing of TLD1433 for treatment of incurable CM and further testing in alternative pre-clinical models.

## 1. Introduction

Conjunctival melanoma (CM) is a rare but often deadly ocular disease that arises from mutated melanocytes, the melanin-producing cells in the conjunctiva [1]. Despite its rare occurrence, conjunctival melanoma is increasing [2]. Current treatment for CM is surgery combined with cryotherapy, brachytherapy, and/or topical chemotherapy. However, efficient treatment for this disease has not yet been developed and the side-effects caused by present treatments are severe [3]. Furthermore, lymph node metastases often develop in spite of a seemingly effective treatment of the local tumor, and may lead to further spreading [4,5,6]. Mutations in BRAF and NRAS genes lead to constitutive activation of the MAPK/ERK signalling pathway, which promotes CM proliferation and survival [6,7,8,9]. 

New treatments for CM are being actively looked for, for example by knocking down the over-expressed gene EZH2 in CM cell lines [10], using e.g., zebrafish xenograft models for faster and more predictive drug testing [11], and by CM-specific gene discovery for targeted therapy [12]. However, cross-talk between different signalling pathways makes specific gene targeting approaches quite challenging [13,14]. In addition, cancer cells proliferate and mutate under single-treatment, often creating resistance against the common cancer drugs or gene-targeting therapies. Considering the superficial growth of CM and the widespread use of laser technology for curing eye diseases, photodynamic therapy (PDT), i.e., tumour ablation through visible light irradiation of a PS located inside the tumour [15,16,17,18], offers an interesting opportunity.

PDT is used for a wide range of dermatological conditions, and for Barrett’s esophagus, bladder and prostate cancer [19]. Compared to surgery, PDT is less invasive, as tumour eradication can be obtained without creating a wound. PDT uses a PS molecule as a prodrug that under light irradiation generates an intense blast of reactive oxygen species (ROS) such as superoxides (O^2−^), hydroxyl radicals (HO^−^), and/or singlet oxygen ^1^O_2_, which eventually cause cytotoxic damage to biomolecules adjacent to the PS. ROS generation inside a malignant tissue may directly destroy cancer cells via ROS-induced cell death, as demonstrated with 5-aminolevulinic acid (5-ALA)-derived protoporphyrin IX (PpIX) [20]. However, indirect anti-tumour effects can also occur via ROS-induced damage to the tumour vasculature, as seen with TOOKAD^®^, mTHPC, or Photofrin^®^ [21,22,23,24,25]. Besides tetrapyrole compounds and their precursors (5-ALA), the metal-based PDT PS, in particular those containing ruthenium(II) (Ru), show a high potential for generating intracellular ROS [26]. These PS absorb in the visible region of the spectrum, which is critical for in vivo activation [27], and suffer less from photobleaching compared to many tetrapyrole compounds [28,29]. 

Recently, the McFarland and Lilge research groups have actively developed TLD1433 (Figure 1a), a Ru-based PS. A phase Ib clinical trial for intravesical PDT treatment of non-muscle invasive bladder tumours using this PS was successfully completed in 2018, [22,29] and a much larger Phase 2 study is underway. The compound has a low cytotoxicity in the dark and an exceptionally high phototoxicity with light activation when tested on a wide range of human cancer cells [29]. Its activation in cells is optimal in the green domain of the spectrum (520 nm), and although the mechanism of cell death by light-activated TLD1433 is not completely clear, this PS generates ^1^O_2_ with near-to-unity quantum efficacy [22]. The outstanding type-II PDT properties of this compound involve initial population of metal-to-ligand charge transfer excited states, which, after intersystem crossing, lead to triplet intra-ligand (3IL) or intra-ligand charge-transfer excited states (3ILCT, Figure 1b) that are very sensitive to oxygen and other excited state quenchers. These remarkable results led us to investigate a possible repurposing of TLD1433 for the PDT treatment of conjunctival melanoma (CM). We analysed the in vitro and in vivo efficacy of this compound in different CM cells lines and zebrafish embryonic xenograft CM models using green light irradiation (λ = 520 nm).

Zebrafish (*Danio rerio*) are indeed increasingly used as an in vivo model to study cancer [30]. Benefits include large clutch size, ex utero development and easy manipulability of larvae [31]. Because there is high conservation of genes between zebrafish (ZF) and human, data collected in ZF are relevant for humans [32]. Notably, the histology of ZF tumours has been shown to be highly similar to tumours found in human cancers [33].

The adaptive immune system in ZF does not reach maturity until 4 weeks post-fertilization, allowing circumvention of cell graft-host rejection by using ZF in early stages [34]. ZF larvae can absorb various small molecular weight compounds from the water they swim in, which is advantageous when screening for anti-cancer compounds. When assessing drug efficacy, ZF experiments require much less material than mouse models [35]. Routinely 1 mL (1 nM to 20 µM) of drug solution is enough for testing drug efficacy in six individual ZF embryos. Alternatively, (pro)drugs can also be injected in the animal in nL quantities, which further minimizes the amount of compound required for testing. Importantly, the use of transgenic lines with fluorescent vasculature, neutrophil granulocytes or macrophages, allows live, non-invasive imaging of proliferation, migration and tumour-associated neo-angiogenesis, and interaction with the microenvironment at the single cell resolution in the entire organism within 1 week [36,37]. For PDT, the transparency of the animals allows for activating a PS in the entire organism by simple light irradiation. Overall, these combined advantages account for the increased experimental use of zebrafish cancer models in drug discovery during the last two decades [38,39]. 

For cutaneous melanoma, a current phase I/II clinical trial of leflunomide combined with vemurafenib is the first to arise from initial screens in zebrafish [40]. Many ZF xenograft models have been established, and the choice of the best ZF model depends on the type of disease, but also on the type of treatment. Human tumour cells can be injected for example into the yolk sac [41], the Duct of Cuvier [42], the pericardial cavity [43], the perivitelline space [44], the swimming bladder [45], or the hindbrain [46]. Here, we aimed to engage different CM xenograft models for testing the TLD1433 PS as a potential new PDT treatment strategy to combat CM growth. We hence developed a new orthotopic model for CM by RO injection of CM cells, mimicking primary tumour spread. We also investigated a previously-developed ectopic model, generated by intravenous cell injection: circulating cancer cells usually form tumour lesions in the tail of the embryo [11,42]. Using three different treatment modalities of TLD1433 in two different tumour models, we established a testing platform in which the anti-tumour efficacy of this PS can be observed.

## 2. Results

### 2.1. TLD1433 Is Phototoxic in Six Eye Melanoma Cell Lines

TLD1433 is known to generate reactive oxygen species (ROS) with high quantum efficacy in many cancer cell lines. However, there is no report of the in vitro toxicity of this compound in eye melanoma (UM) cell lines. We determined the cell viability of three conjunctival melanoma cell lines (CRMM1, CRMM2 and CM2005.1), and three uveal melanoma cell lines (OMM1, OMM2.5, MEL270) in the presence of TLD1433, both in the dark or under green light irradiation (21 mW/cm^2^, 19 J/cm^2^, 520 nm, 15 min), and compared this viability to epidermoid carcinoma A431 and cutaneous melanoma A375 cell lines under the same conditions. The protocol used was based on previous work from the Bonnet group [47,48], and differed slightly from recommendations from McFarland et al. [19]. Notably, the cell seeding time was 24 h instead of 3 h, and the drug-to-light interval (DLI) was 24 h instead of 16 h. The effective concentration (EC50) values, i.e., the concentration required to reduce cell viability by 50% compared to untreated wells, were assessed by fitting the dose-response curves with a Hill equation. The phototoxicity index (PI), defined as the ratio of the dark EC50 to the light EC50, was also calculated and represents the amplification of TLD1433 activity with a light trigger. In control A375 and A431 cells, the dark toxicity of TLD1433 was very low, with EC50 values higher than the highest concentration used in that assay (5 µM). PI is greater than 100, as previously reported for other cell lines [22], were observed. In eye melanoma cells, the dark toxicity of TLD1433 was relatively high, with EC50 values around 1 µM. Upon light activation, TLD1433 became significantly more potent (as observed with the A375 and A431 cell lines), with EC50 values in the nanomolar regime. The lowest EC50 values were measured for CM cells, where the PI values were also the highest (>140). Due to the higher PI values for CM cells compared to uveal or cutaneous cancer cells, CM cells were chosen for later in vivo experiments (Figure 2 and Table 1). 

It should be noted that the PIs determined were somewhat lower than those reported for TLD1433 with other cell lines (>1000), which could be due either to a preferential toxicity toward uveal and CM melanoma lines or a difference in the in vitro PDT protocol used or both. Under the selected conditions, the dark toxicity observed for both uveal and CM cell lines was relatively high [29], which reduces the maximum PIs that can be obtained. In addition, the slightly lower PI could be due to the low light dose that we used, compared to other studies: typically, 100 J.cm^−2^ has been proposed by McFarland et al. [19]. Regardless, we chose CM cells for further studies given that TLD1433 had the largest PI and was most phototoxic toward this particular cell line.

### 2.2. TLD1433 Induces Apoptosis and Necrosis in CRMM1 and CRMM2 Cells

Depending on the nature and intracellular localization of a PS, the light dose, and the cell type, PDT is known to provoke either necrosis, apoptosis, or autophagy [49]. In order to investigate the death mechanism induced by green light-activated TLD1433 in CRMM1 and CRMM2 cells, the cells were stained with Annexin V and propidium iodide, and further analysed by fluorescence-activated cell sorting (FACS). 

In both the vehicle control and the dark TLD1433 groups, most cells were found alive. In the light-activated TLD1433 group, about half of the cells were found dead, either in the late apoptotic or necrotic quadrant (Figure 3A–D), but most importantly, very few early apoptotic cells were found. Overall, these results suggest that the CM cells treated with TLD1433 and light did not die via apoptosis, but probably by necrosis.

### 2.3. Light Toxicity and the Maximum Tolerated Dose of TLD1433 by Water, Intra-Venous and RO Administration in Zebrafish Embryos

In order to test the effectiveness of TLD1433-induced PDT in zebrafish CM cancer models, we first examined the effect of light irradiation on wild type, non-injected embryos. 2 days post fertilization (dpf) embryos (30 embryos per group) were exposed continuously to green light (21 mW/cm^2^, 520 nm) for 0, 3, 6, and 12 h and cytotoxicity symptoms were monitored by stereomicroscope. 

Green light irradiation for 6 h did not induce any toxicity or developmental defects in zebrafish embryos; the percentage of mortality, malformation (i.e., bent spine and pericardial edema) and fish length were the same as in the control group (Figure 4), indicating that green light is not toxic to ZF until at least 6 h of continued exposure. 

Next, we tried three different regimens of drug administration into zebrafish larvae and determined the maximum tolerated dose (MTD) of TLD1433 in dark and after light activation (Figure 5 and Table 2). Water administration (WA) of drugs by skin epithelial cell absorption and drinking is commonly used in zebrafish drug experiments [35]. Hence, different concentrations of TLD1433 were added into the egg water at 2.5, 3.5, 4.5, and 5.5 dpf embryos, followed by 12 h DLI and 90 min green light irradiation (21 mW/cm^2^, 114 J.cm^−2^, 520 nm). In addition, we also tested IV of TLD1433 by direct injection into the dorsal vein, as well behind the eye injections for RO administration [41]. For IV and RO administration, the compound was injected four times into the embryos at 3, 4, 5 and 6dpf, followed by 60 min drug-to-light interval and the same kinetic and irradiation regime as for the WA administration (Figure 5A). Zebrafish embryos tolerated light-activated TLD1433 without any effect on the mortality, malformation and fish length at an MTD of 9.2 nM when delivered by WA administration and an MTD of 4.6mM when delivered by IV and RO administration, respectively (Figure 5B–D). Considering that in the dark, even higher concentrations of TLD1433 (23 nM by WA, 11.5 mM by IV and RO) were not toxic to embryos, we conclude that this compound is activated by green light irradiation and very effective at low concentrations in vivo.

### 2.4. The Treatment of TLD1433 by WA, IV and RO in a Zebrafish Ectopic and Orthotopic Tumour Model

The zebrafish ectopic conjunctival melanoma tumour model has been described previously [42]. In this model, around 200 fluorescent CM cells are injected into the Duct of Cuvier at 2 dpf, and then disseminate through the blood circulation and grow in the head and tail. To establish the orthotopic tumour model, around 100 red, (td)Tomato fluorescent CRMM1 or CRMM2 cells were injected RO at 2 dpf into tg(Fli:GFP/Casper), endothelial reporter transgenic zebrafish with green fluorescent vasculature and examined by fluorescent microscopy at day 1 and 4 after engraftment (Figure 6A). Tumour expansion at the injection site was measured as fluorescence intensity and tumour area. Figure 6B shows that RO-engrafted CRMM1 and CRMM2 significantly proliferated at the site of injection and formed primary tumour lesions (Figure 6B,C). 

To engage both CM models for testing the efficacy of TLD1433 as a potential new PDT treatment strategy to combat CM growth, first, the MTD of TLD1433 delivered into zebrafish embryos engrafted with CM cells was measured (Table 2) following the same procedure as already described for wild type embryos (Figure 5 and Table 2). Engrafted embryos were more sensitive to light-activated TLD1433 than non-engrafted embryos (Table 2). MTD concentrations of 4.6 nM and 2.3 mM were delivered by WA, IV and RO administration. Delivery of TLD1433 at the MTD by WA did not inhibit tumour burden in the ectopic or orthotopic tumour model after engraftment of CRMM1 and CRMM2 cells (Figure 7). Relative tumour burden, estimated as fluorescence intensity and tumour area, was not significantly different between the dark and light treatments (21 mW/cm^2^, 114 J.cm^−2^, 520 nm), indicating that the low concentrations of TLD1433 added to the water of engrafted embryos were not sufficient to attenuate CM growth in either model (Table 3 and Table 4). The TLD1433 concentration in these experiments was not increased further as the initial treatment was already at the pre-determined MTD. 

Next, the effect of IV administration of TLD1433 was determined in both CM models. Figure 8 indicates that light activation of TLD1433 significantly reduced the tumour burden in the CRMM1 and CRMM2 ectopic model but not in the orthotopic model. In the ectopic model, light activation with the MTD (2.3 mM) of TLD1433 (21 mW/cm^2^, 114 J.cm^−2^, 520 nm) inhibited the CRMM1 and CRMM2 tumour fluorescence intensity as well as tumour area (41%, 31%, 54%, 50%) (Figure 8B,C and Table 3 and Table 4). The CRMM1 and CRMM2 tumour burden was not changed in the orthotopic model (Figure 8D,E and Table 3 and Table 4). This clearly shows that CRMM1 and CRMM2 tumour cells received a sufficient amount of activated TLD1433 in the ectopic model but not in the orthotopic model, suggesting that IV administration allows the compound to reach and inhibit CM cells in the ectopic model but is not effective to attenuate localized CM growth behind the eye in orthotopic model.

In contrast, delivery of the same concentration of TLD1433 (2.3 mM) by RO administration toward CRMM1 and CRMM2-induced tumours diminished the fluorescence intensity and tumour area in both ectopic (47%, 40%, 64%, 52%) and orthotopic models (35%, 55%, 69%, 71%) upon green light activation (114 J.cm^−2^, 520 nm) (Figure 9 and Table 3 and Table 4). We propose that TLD1433 remained longer in the interstitial fluid at the injection site after RO injection, reaching a higher effective concentration to inhibit CM cells grown in the same area. 

### 2.5. TLD1433 by Retro Orbital Administration Induces Apoptosis of CRMM1 and CRMM2 Cells in Zebrafish Orthotopic Model

In situ TUNEL assay on fixed embryos was used to detect TLD1433 induced apoptosis in zebrafish CRMM1 and CRMM2 orthotopic tumour models at 4 dpi after light activation of 2.3 mM TLD1433, administrated by retro orbital injection. The DNA strand breaks in apoptotic tumour cells were stained with fluorescein and visualized as a green signal. In control dark, control light, TLD1433 dark groups there was no positive green signal detected, which co-localized with red signal of CRMM1 and CRMM2 engrafted cells (Figure 10). In contrast, light activation of TLD1433, as described before (Figure 9C,E), induced CRMM1 and CRMM2 cell apoptosis in the zebrafish orthotopic model. After light irradiation, the red signal representing engrafted CM cells was reduced, however some of the remaining cells stained positive for apoptotic cells and turned green (yellow in overlay), indicating that PDT-driven anti-tumour efficacy of TLD1433 in this PDT regimen is at least partially apoptosis-dependent. 

## 3. Discussion

Developing new ocular PDT treatments often depends on a limited number of rabbit studies, due to lack of other animal models. To overcome this, we previously generated an ectopic CM model, and now developed an orthotopic CM model in zebrafish. Zebrafish xenograft models are particularly straightforward for testing compound toxicity and efficacy in vivo, as due to the small size and transparence of the embryo, one can examine on the one hand adverse effects on developing phenotypes or animal survival, and on the other hand tumour burden by fluorescence microscopy. For PDT in zebrafish, one should note that the PI can either be defined as the total tumour fluorescence or the total tumour area (as detected in confocal microscopy) in the light-activated group, divided by the tumour fluorescence or tumour area in the dark group. These definitions are quite different from the definition of the PI in vitro, where it is usually defined as the ratio between the EC_50_ values in the dark and in light-irradiated conditions. As a consequence, in vitro and in vivo PIs cannot be directly compared. For example, the PI obtained by fluorescence spectroscopy in the orthotopic CRMM1 model and using RO injection of TLD1433 was 1.85, while that obtained by measuring the tumour area was 4.1; the PI measured by the ratio of EC_50_ values in vitro was 140. The only PIs that can be compared are the ones defined identically in the same cancer model. 

Here, our results demonstrate not only activity of TLD1433 in a broad range of different CM and UM cells in vitro, but also anti-tumour activity in a zebrafish embryo tumour models of CM. Interestingly, the in vitro results on 6 different eye melanoma cell lines are not significantly different, which means that TLD1433 shows a broad range of photoactivity, independently of the genetic background of the different cell lines. For the in vivo part of this work, we focussed on CM because TLD1433 induced the highest PIs in these cell lines. However, future experiments may further analyse UM, as good activity was also observed in the UM cell lines. Clearly, the excellent photodynamic properties of the Ru-based TLD1433 sensitizer make it phototoxic in most cell lines, including cutaneous melanoma and non-melanoma cell lines. When testing it in vivo, it is hence particularly important to optimise the mode of administration, compound dose, and light dose, in order to minimise side effects.

In ZF embryo models for PDT the small size of the embryo, of the tumor, and the relative optical transparency of all tissues, combine into easy light penetration into the tumor. On the other hand, local irradiation of the tumor is very difficult, so that it was not investigated here. Hence, the main challenge in developing a ZF embryo model for testing PDT sensitizers, is to insure that the concentration of the photosensitizer in the tumor tissue is, at the moment of irradiation, high enough, while it should remain as low as possible in healthy tissues, which will be irradiated as well. This condition is the only way to insure, following light irradiation, a maximum dose of oxidative stress in the diseased tissue, while keeping minimal the activity in the rest of the body. In larger animals such as mice or humans, achieving deep light penetration in the tumor tissue and constant light dosimetry can be tougher and requires specific optimization for each type of cancer, compound, and irradiation wavelength. On the other hand, light irradiation is, by definition of PDT, circumvented to the tumor area, which minimizes global toxicity issues. However, in this stage of our research we cannot exclude an inflammatory reaction caused exclusively by innate immunity cells (as adaptive immunity is not active yet) of the zebrafish embryos elicited by tumour necrosis or apoptosis after TLD treatment. This reaction may drive same negative side effects or even help to attenuate tumour development in this model [50].

Whether the transparency of ZF embryo is considered as an advantage or a disadvantage for testing PDT photosensitizers, the relationship between the method of implanting the tumour and the mode of administration of the compound has to be established for each particular disease. For CM, our results clearly demonstrate that when the route of administration did not fit with the chosen tumour model, the activity in zebrafish was abrogated. This result is very important considering the notoriously excellent ROS generation properties of TLD1433 and its excellent PDT properties in mice tumour models [19]. In other words, activity in fish only appeared when the proper administration route was used. Independently of the model, water administration did not to allow for the compound to reach the tumor. For intraveinous injection of TLD1433, the situation was more contrasted, as the ectopic model showed good activity after light irradiation, while the orthotopic model did not. In the ectopic model, engrafted cells disseminate through the blood circulation and form small metastatic lesions in the head and tail. At this stage, we can only speculate that when TLD1433 is also injected in the blood circulation more cells may effectively be reached by the drug before activation. In the orthotopic model, lesions are bigger and may be more compact, so that cells may be less easily reached by the drug when it is injected in blood. When TLD1433 was locally injected behind the eye, an excellent response was found for both ectopic and orthotopic models. Overall, the best response was obtained with the orthotopic model, combined with local injection of TLD1433, i.e., injection behind the eye. In such a case, maximum drug concentration may be achieved in the tumor right before light irradiation. Such a mode of administration turns out to be reminiscent of that used in bladder cancer patient, where TLD1433 is injected in the bladder and taken up very selectively by the tumour cells. Our results open the door for further zebrafish testing of not only TLD1433 (to assess on its toxicity, activity, and mode of action), but also of other phototherapeutic compounds, for which no activity in vivo has ever been reported. 

Last but not least, clinical PDT in intraocular melanoma has up to now been limited not only by the lack of clinically approved PS, but also by interferences by the ocular and tumour pigment with light absorption. Most approved PDT sensitizers are porphyrin compounds, which offer a quite narrow (~20 nm) excitation wavelength range. If the pigment of the tumour absorbs too much of that light, PDT activity may be compromised. TLD1433, like most Ru polypyridyl compounds, shows broad absorption bands (Δλ ~ 150 nm) between the blue and red regions of the spectrum, thereby allowing to fine-tune the excitation wavelength and optimise light absorption by the sensitizer vs. light absorption by the pigment [51]. These effects could not be tested here, as the CM and uveal melanoma cell lines have lost their pigments. Rutherrin, a new formulation of TLD1433 and transferrin, is now being proposed to improve the target specificity and water solubility of the PS [52,53,54]. Rutherrin was proven to cross the blood brain barrier (BBB) and is now under clinical investigation for glioblastoma multiforme (GBM) and non-small-cell lung cancer (NSCLC) [19]. However, this obstacle should be taken into account in any further in vivo testing of Ru-based sensitizers for PDT.

## 4. Materials and Methods 

### 4.1. Photosensitizers

For in vitro studies, TLD1433 was firstly diluted to 2 mM in autoclaved PBS and further diluted in media as required. For the in vivo studies, TLD1433 was directly diluted to autoclaved 2% PVP as required.

### 4.2. Culturing Cell Lines

Human conjunctival malignant melanoma cell lines CRMM1 and CRMM2, isolated by Nareyeck et al. [55], were cultured in F12 Kaighn’s modified medium (cat# SH30526.01, Hyclone, South Logan, UT, USA) supplemented with 10% fetal bovine serum (FBS; Gibco, Carlsbad, CA, USA). CM2005.1 established by Keijser et al. [56] was cultured in RPMI 1640, Dutch Modified (cat# 22409-015, Life Technologies, Carlsbad, CA, USA), supplemented with 10% fetal bovine serum (FBS; Gibco), 3 mM L-glutamine (1%, cat# 35050-038, Life Technologies). Human uveal melanoma cell lines OMM1 (provided by Prof. Dr. G.P.M Luyten) [57], OMM2.5, MEL270 (provided by Dr. B.R. Ksander BR) [58]) were cultured in Ham’s F12 medium (cat# N3790, Sigma-Aldrich, Zwijndrecht, Netherlands) supplemented with 10% FCS. Stable fluorescent CRMM1 and CRMM2 cell lines were generated using lentivirus expressing both tandem dimer (td) tomato and blasticidin-S, as previously described [59]. Human cancer cell lines A431 and A375 were distributed by the European Collection of Cell Cultures (ECACC), and purchased through Sigma Aldrich (Zwijndrecht, Netherlands). A375 and A431 cells were thawed and at least passaged twice before starting cytotoxicity and uptake experiments. A375 and A431 cells were cultured in Dulbecco’s Modified Eagle Medium with phenol red, supplemented with 10.0% *v*/*v* FCS, 0.2% *v*/*v* penicillin/streptomycin (P/S), and 0.9% *v*/*v* glutamine-S (GM). Cells were cultured in either 25 cm^2^ or 75 cm^2^ flasks and split at 70–80% confluence. The flasks were incubated in a normoxic incubator at 37 °C at 5.0% CO^2^ in a PHCbi O^2^/CO^2^ incubator, MCO-170M). The medium was refreshed twice a week. Cells used in all biological experiments were cultured for not more than eight weeks. Dulbecco’s Minimal Essential Medium (DMEM, high glucose, without glutamine), 200 mM glutamine-S (GM), trichloroacetic acid (TCA), glacial acetic acid, sulforhodamine B (SRB), tris (hydroxylmethyl) aminomethane (tris base), and *cis*-diamineplatinum (II) dichloride (cisplatin) were purchased from Sigma Aldrich. (2R,3R,4R,5R)-hexan-1,2,3,4,5,6-hexol (D-mannitol) was purchased from Santa Cruz Biotechnology via Bio-Connect (Huissen, Netherlands). FCS was purchased from Hyclone. Penicillin and streptomycin were purchased from Duchefa (Haarlem, Netherlands) and were diluted to a 100 mg/mL penicillin/streptomycin solution (P/S). Trypsin and Opti-MEM^®^ (without phenol red) were purchased from Gibco^®^ Life Technologies (Carlsbad, CA, USA). Trypan blue (0.4% in 0.81% sodium chloride and 0.06% potassium phosphate dibasic solution) was purchased from Bio-Rad (Gelderland, The Netherlands). Plastic disposable flasks and 96-well plates were obtained from Sarstedt (Amsterdam, Netherlands). Cells were counted using a Bio-Rad TC10 automated cell counter with Bio-Rad Cell Counting Slides.

### 4.3. In Vitro Cytotoxicity (SRB) Assay

At day 0, cells were detached using 1mL of trypsin, resuspended in 4 mL of media and transferred to a 15 mL corning falcon tube. Cells were counted using trypan blue and BioRad^®^ TC20™ automated cell counter (Figure 11). Dilutions of 6000 (CRMM1), 6000 (CRMM2), 8000 (CM2005.1), 6000 (OMM1), 6000 (OMM2.5), 6000 (MEL270) 8000 (A431), and 4000 (A375) cells/well were calculated from each cell suspension at a final volume of 6 mL using the following formula:
$$V_{c}=(V_{t}\times 10C)/L_{c}$$
where *V_t_* = total volume of solution, *C* = number of cells per well/per 100 μL and *L_c_* = live cells count (cells/mL). The cell suspensions were transferred to a 50 mL reservoir and 100 µl of each cell line was seeded at the aforementioned cell densities in triplicate in six 96-well plates. Boarder wells were intentionally filled with PBS media to avoid boarder effects. After 24 h, the cells were treated with TLD1433 with six different concentrations ranging from 0.025 µM to 3.0 µM, followed by incubation in a normoxic incubator. After 24 h of post treatment the cells were exposed to the green light for 15 min (520 nm, 21 mW/cm^2^, 19 J/cm^2^). The dark control plate was kept under dark conditions. Cisplatin was used as a positive control in all cell types. Then cells were incubated for another 48 h before fixing them with trichloroacetic acid (10% *w*/*w*) solution. The fixed cells were kept at 4 ℃ for 48 h, when TCA was washed out with distilled water before adding the sulphorhodamin B (SRB) (0.6% SRB) dye. The SRB dye was washed out after 30 min and plates were air dried for overnight. Next day, the dye was dissolved using Tri-base (0.25%) and absorbance of SRB at 510 nm was recorded from each well using a Tecan plate reader. The SRB absorbance data was used to calculate the fraction of viable cells in each well (Excel and GraphPad Prism software). The absorbance data were averaged from triplicate wells per concentration. Relative cell viabilities were calculated by dividing the average absorbance of the treated wells by the average absorbance of the untreated wells. Three independent biological replicates were completed for each cell line (three different passage numbers per cell line). The average cell viability of the three biological replicates was plotted versus log(concentration) [μM], with the SD error of each point. By using the dose–response curve for each cell line under dark- and irradiated conditions, the effective concentration (EC_50_) was calculated by fitting the curves to a non-linear regression function with a fixed maximum (100%) and minimum (0%) (relative cell viability) and a variable Hill slope, which resulted in the simplified two-parameter Hill-slope equation [Equation (1)]:100/(1 + 10((log10EC_50_ − X)⋅HillSlope(1)

### 4.4. Cell Irradiation Setup

The cell irradiation system consisted of a Ditabis thermostat (980923001, Ditabis Digital Biomedical, Pforzheim, Deutscheland) fitted with two flat-bottomed microplate thermoblocks (800010600) and a 96-LED array fitted to a standard 96-well plate. The λ = 520 nm LED (OVL3324), fans (40 mm, 24 V DC, 9714839), and power supply (EA PS 2042-06B) were obtained from Farnell (Leeds, England) as reported in our previous publication [47].

### 4.5. Flow Cytometry

CRMM1 (10,000/well) and CRMM2 (10,000/well) cells were seeded into an 8-well chamber in Opti-MEMTM (Gibco, Reduced Serum Medium, no phenol red) with 2.5% FBS (Gibco). After 24 h incubation, TLD1433 (0.0059 μM for CRMM1, 0.0048 μM for CRMM2) was added into the medium. 24 h later, wells were washed and new drug-free medium was added. The cells were exposed to green light (520 nm, 21 mW/cm^2^, 19 J/cm^2^) for 15 min and incubated for 48h. Medium of all wells was collected and wells were washed with PBS and lysed by 500 µL trypsin for 3 min. Collected medium was added to the wells with lysed cells, mixed and centrifuged for 2000 rpm, 3 min. After washing, cells were resuspended in 200 µL of 1× binding buffer. Next, 5 µL of Annexin-V-FITC and 5 µL of Propidium Iodide was added to each well and incubated for 15 min at room temperature. 200 µL of sample was added to 96-well plate, and used for FACS measurement.

### 4.6. Zebrafish Maintenance, Tumour Cells Implantation and Tumour Analysis 

Zebrafish lines were kept in compliance with the local animal welfare regulations and European directives. The study was approved by the local animal welfare committee (DEC) of the University of Leiden (Project: “Anticancer compound and target discovery in zebrafish xenograft model”. License number: AVD1060020172410). The Zebrafish (ZF) Tg(fli1: GFP/Casper) [36] were handled in compliance with local animal welfare regulations and maintained according to standard protocols (www.ZFIN.org).

For cancer cell injection, two days post-fertilization (dpf), dechorionated zebrafish embryos were anaesthetized with 0.003% tricaine (Sigma) and plated on a 10 cm Petri dish covered with 1.5% of solidified agarose. CRMM1 and CRMM2 cells were suspended in PBS containing 2% polyvinylpyrrolidone (PVP; Sigma-Aldrich) with a concentration of 50,000 cells/µL and loaded into borosilicate glass capillary needles (1 mm O.D. × 0.78 mm I.D.; Harvard Apparatus). In the ectopic model, 200 (td)Tomato fluorescent CM cells were injected into the Duct of Cuvier or at 2 dpf, which led to dissemination through the blood circulation and outgrowth in the head and tail. In orthotopic tumour model, 100 (td)Tomato fluorescent CRMM1 or CRMM2 cells were injected RO in 2 dpf embryos using a Pneumatic Picopump and a manipulator (WPI). After injection, the embryos were incubated in a 34 °C incubator. Images were acquired at 1-, 2- and 4-days post injection (dpi) with a Leica M165 FC stereo fluorescence microscope. Tumor growth was quantified by calculating the total fluorescence intensity and area with the ZF4 pixel counting program (Leiden). Each experiment was performed at least 3 times with a group size of >30 embryos. 

### 4.7. Light Toxicity Assay for Zebrafish Embryos

Two dpf embryos were transferred into 6-well plates (10 embryos/well). The embryos were exposed to green light (21 mW/cm^2^, 520 nm) for 0, 3, 6, 12 h. After irradiation, images were taken using a DFC420C camera coupled to a MZ16FA fluorescence microscope (Leica, Heerbrugg, Switzerland).

### 4.8. Maximum Tolerated Dose (MTD) for Wild Type Zebrafish and Tumour Cells Injected Zebrafish

For determining the MTD of the WA of the TLD1433 solution in wild type zebrafish, solutions of 2.3 nM, 4.6 nM, 9.2 nM, 11.5 nM, 23 nM were made before the experiment. At 2.5, 3.5, 4.5, 5.5 dpf, TLD1433 was added to the fish water and maintained for 12 h. At 3, 4, 5, 6 dpf, the fish water was refreshed and after 1 h, embryos were exposed to green light for 90 min (520 nm, 21 mW/cm^2^, 114 J/cm^2^). For the IV and RO administration, TLD1433 solutions (1.15 mM, 2.3 mM, 4.6 mM, 9.2 mM, 11.5 mM) were prepared before the experiment. At 3, 4, 5, 6 dpf, 1nL of TLD1433 was injected via the dorsal vein or the RO site and maintained for 1 h. After each of the 4 injections, the embryos were exposed to green light for 90 min (520 nm, 21 mW/cm^2^, 114 J/cm^2^). The images of treated and wild type embryos at 6 dpf were taken using a DFC420C camera coupled to a Leica MZ16FA fluorescence microscope. In order to determine the MTD of tumour cell bearing zebrafish, TLD1433 was performed according to the same procedure, delivered by WA, IV and RA administration as described above for the wild type embryos.

### 4.9. The Efficacy of TLD1433 by WA, IV and RO in a Zebrafish Ectopic and Orthotopic Tumour Models

Fluorescent CRMM1 and CRMM2 cells were injected at 2 dpf into the Duct of Cuvier (ectopic model) and behind the eye (orthotopic model) and TLD1433 was delivered by WA, IV and RO administration with or without a light treatment as described in 4.8. For the WA administration, the 4.6 nM TLD1433 solution was added to the tumour cells injected zebrafish at 2.5, 3.5, 4.5, 5.5 dpf and maintained for 12 h. At 3, 4, 5, 6 dpf, the fish water was refreshed, and after 1 h, embryos were exposed to green light for 90 min (21 mW/cm^2^, 114 J/cm^2^, 520 nm). For the IV and RO administration, 1 nL of 2.3 mM TLD1433 solution was injected via the dorsal vein or the RO site at 3, 4, 5, 6 dpf. After 1 h interval, the embryos were exposed to green light for 90 min (21 mW/cm^2^, 114 J/cm^2^, 520 nm). After treatment, the embryos images were acquired with a Leica M165 FC stereo fluorescence microscope. Tumor growth was quantified by calculating the total fluorescence intensity and area with the ZF4 pixel counting program (Leiden). Each experiment was performed at least 3 times with a group size of >30 embryos.

### 4.10. TUNEL Assay

The zebrafish larvae were fixed overnight with 4% PFA at 4 °C. Embryos were washed in PBST for 5 min and dehydrated by a graded methanol series until reaching 100% methanol. Embryos were stored at −20 °C for further use. Embryos were gradually rehydrated in PBST (25%, 50%, 75%), washed twice for 10 min with PBST and digested by proteinase K (Roche, Mannheim, Germany) solution in PBST (10 µg/mL) at 37 °C for 40 min. After two washes in PBST, embryos were post-fixed in 4% PFA for 20 min. After twice washing in PBST for 10 min, 50 µL of TdT reaction mix (Roche) was added to the embryos. Embryos were overnight incubated with the TdT at 37 °C (in the dark). The reaction was stopped by three 15 min washes with PBST at room temperature and embryos were used for high-resolution imaging. Embryos were placed on glass-bottom petri dishes and covered with 1% low melting agarose containing 0.003% tricaine (Sigma). Imaging was performed using the Leica SP8 confocal microscope. The images were processed with ImageJ software (National Institutes of Health, Bethesda, MD, USA). Each experiment was performed three times with a group size of 10 embryos. 

### 4.11. Statistical Analysis

Determination of the EC50 concentrations in vitro was based on a non-linear regression analysis performed using GraphPad Prism Software. Results are presented as means ± SD from three independent experiments. Significant differences were detected by one-way ANOVA followed by Dunnett’s multiple comparisons test implemented by Prism 8 (GraphPad Software Inc., La Jolla, CA, USA). A *p*-value < 0.05 was considered statistically significant (**p* ≤ 0.05, ** *p* ≤ 0.01, *** *p* ≤ 0.001, **** *p* ≤ 0.0001).

## 5. Conclusions

Our work supports three main conclusions. First, the Ru-based PDT sensitizer TLD1433 is very active in eye melanoma cell lines, where green light activation provokes cell death via apoptosis and necrosis. Second, this paper is one of the rare examples of testing PDT in a zebrafish tumour model. It could hence act as a basis for future PDT sensitizer screening in vivo, somewhere between in vitro and mice studies. Due to the excellent ROS generation properties of this PDT sensitizer, it appears of utmost importance to fine-tune the way of administration of the prodrug to the tumour model. For two different models of conjunctive melanoma, i.e., an ectopic and an orthotopic model, we have tested three ways of administration of TLD1433. The WA, which is often chosen to test compounds in zebrafish, did not give good results: the phototoxicity to the zebrafish was high, and the anti-tumour efficacy low. When the compound was injected IV or RO, however, the toxicity became much lower and, when injected IV or RO, excellent anti-tumour properties were observed. We hence propose, as a third and last conclusion of this work, that TLD1433 can be repurposed as a treatment against conjunctival melanoma.

## Figures and Tables

**Figure 1 cancers-12-00587-f001:**
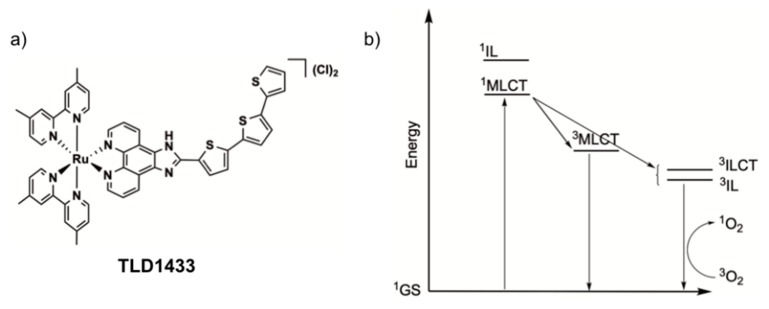
(**a**) Chemical structure of the PDT photosynthesizer TLD1433. (**b**) Jablonski diagram showing the formation of singlet oxygen (^1^O_2_) by irradiation of TLD1433 via initial population of metal-to-ligand charge transfer (MLCT) states and intersystem crossing to intra-ligand (IL, ILCT) states.

**Figure 2 cancers-12-00587-f002:**
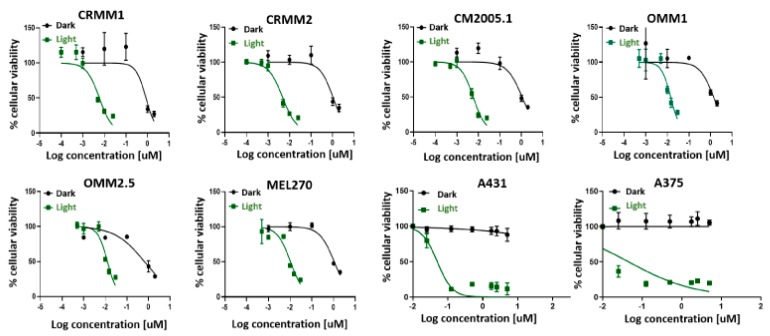
Cell viability after TLD1433 treatment of eight tumour cell lines (CRMM1, CRMM2, CM2005.1, OMM1, OMM2.5, MEL270, A431, A375). The green line shows TLD1433 activated by 520 nm light, 21 mW/cm^2^, 19 J.cm^−2^ (light-induced toxicity). The dark line shows TLD1433 treatment without light irradiation (dark toxicity). The tumour cells were treated with TLD1433 for 24 h with concentrations ranging from 0.001 µM to 5 µM and kept in the dark, or ranging from 0.0001 µM to 0.025 µM and illuminated with a light dose of 21 mW/cm^2^, 19 J/cm^2^. SRB assay was carried out at 48 h after light irradiation. The absorbance of Sulforhodamine B in solution was measured at 520 nm. Results are presented as means ± SD from three independent experiments with 95% confidence intervals.

**Figure 3 cancers-12-00587-f003:**
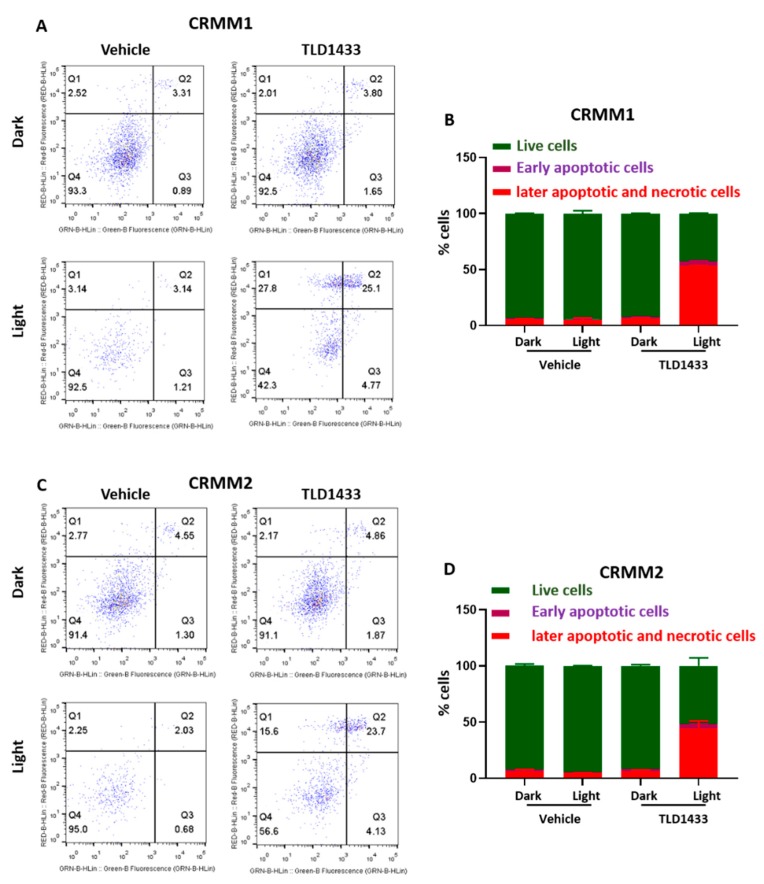
Green light irradiation of TLD1433 induces apoptosis and necrosis in CRMM1 and CRMM2 cells. (**A**) CRMM1 and (**C**) CRMM2 were stained with Annexin-V-FITC and Propidium Iodide. The percentages of live, early apoptotic, later apoptotic and necrotic cells in CRMM1 (**B**) and CRMM2 (**D**) were counted by FACS. Results are presented as means ± SD from three independent experiments.

**Figure 4 cancers-12-00587-f004:**
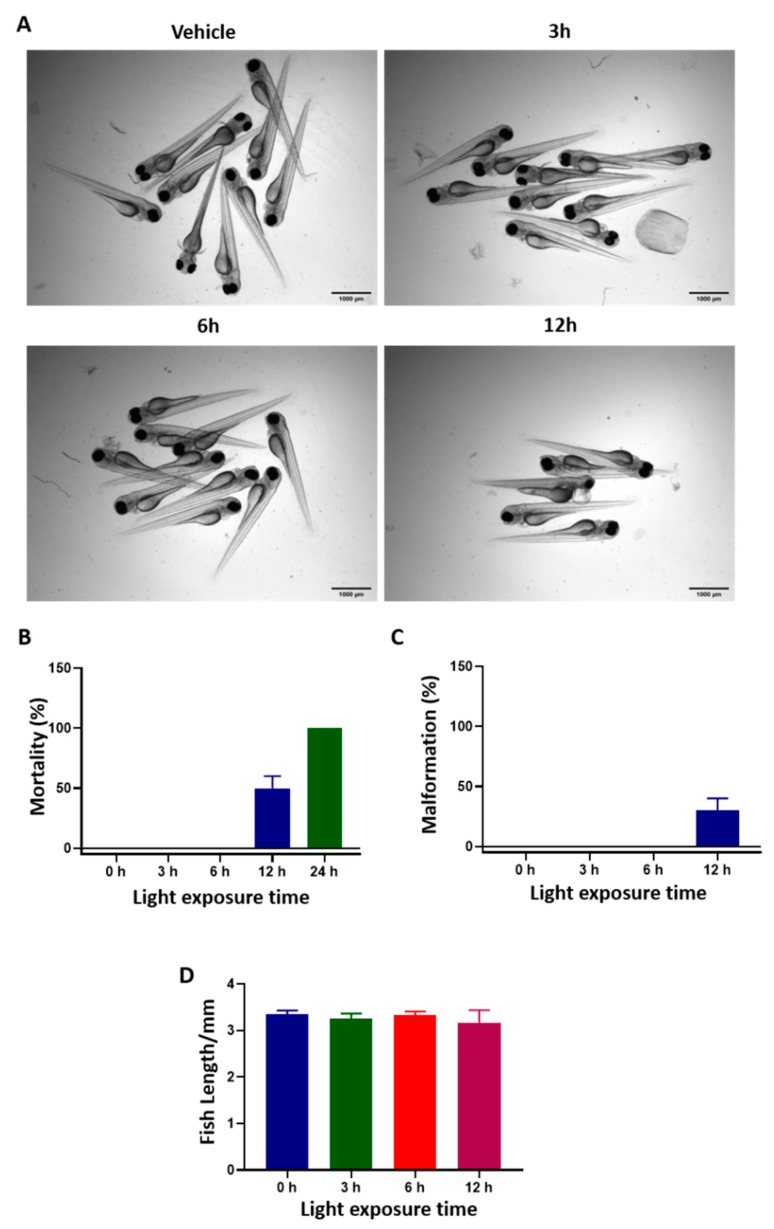
Light toxicity in zebrafish embryos. 2 dpf embryos (*n* = 30) were exposed to green light (21 mW/cm^2^, 520 nm) for 0, 3, 6, or 12 h. (**A**) Transmitted light images of the embryos after light irradiation. (**B**–**D**) The percentage of mortality, malformation and fish length after various time of light exposure. Results represents the means ± SD from three independent experiments.

**Figure 5 cancers-12-00587-f005:**
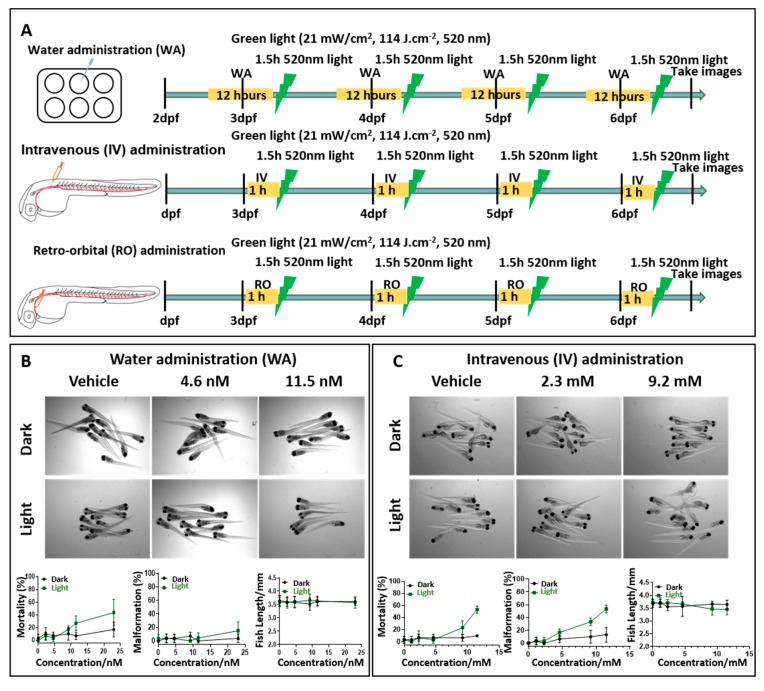
The maximum tolerated dose of TLD1433 in wild type zebrafish embryos administered through three different routes. (**A**) Schedule of TLD1433 treatment in wild type zebrafish. WA: TLD1433 (2.3 nM, 4.6 nM, 9.2 nM, 11.5 nM, 23 nM) were added to the water containing 10 embryos per well at 2.5, 3.5, 4.5, 5.5 dpf, for 12 h (yellow box). After these treatments, the drug was removed and replaced by egg water followed by 90 min green light irradiation (21 mW/cm^2^, 114J.cm^−2^, 520 nm), depicted as a green lightning bolt. IV or RO: 1 nL of TLD1433 (1.15 mM, 2.3 mM, 4.6 mM, 9.2 mM, 11.5 mM) were injected into the embryos at 3 dpf to 6 dpf every morning, followed by 60 min drug-to-light interval (yellow box) and 90 min green light irradiation (21 mW/cm^2^, 114 J.cm^−2^, 520 nm), depicted as a green lightning bolt. (**B**) WA, (**C**) IV, (**D**) RO. (**B**–**D**) Images were made of irradiated (light) and non-irradiated (dark) embryos (*n* = 30) at 6dpf and the percentages of mortality, malformation and fish length were calculated (shown as means ± SD from three independent experiments). Representative images of embryos under dark and light conditions are shown.

**Figure 6 cancers-12-00587-f006:**
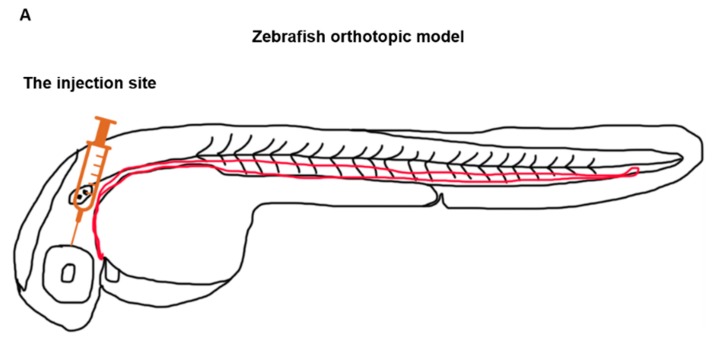
Development of a new conjunctival melanoma orthotopic tumour model in ZF. (**A**) Location of CM cell injection, (**B**) red fluorescent CRMM1 and (C) CRMM2 cells were injected RO into 2 dpf tg(Fli:GFP/Casper) (*n* = 10) and imaged by fluorescence microscopy at 1 and 4 days post injection (dpi). Relative tumour burden was calculated as fluorescent intensity and area of tumour cells by Image J. Results are means ± SD of three independent experiments.

**Figure 7 cancers-12-00587-f007:**
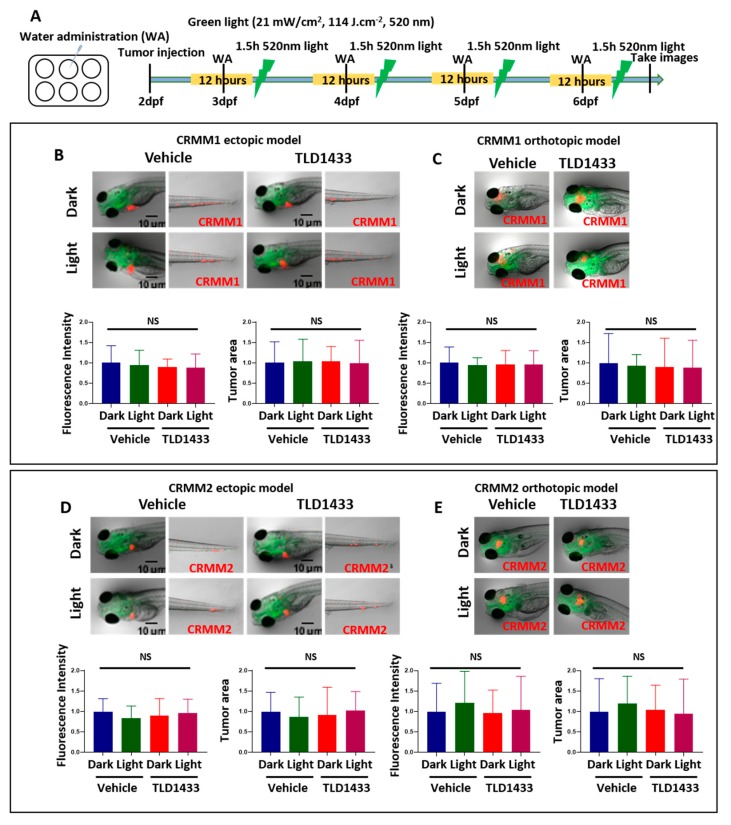
Treatment of zebrafish ectopic and orthotopic CM models with TLD1433 through WA. (**A**) Schedule of tumour injection and TLD1433 administration in zebrafish embryos. Fluorescent CRMM1 and CRMM2 cells were injected at 2 dpf into the Duct of Cuvier (ectopic model) and behind the eye (orthotopic model) and TLD1433 was administered with or without a light treatment following the schedule presented in A. Relative tumour burden estimated as fluorescence intensity and tumour area was calculated by Image J. (**B**) CRMM1 tumour burden in ectopic model (*n* ≈ 30). (**C**) CRMM1 tumour burden in orthotopic model (*n* ≈ 15). (D) CRMM2 tumour burden in ectopic model (*n* ≈ 30). (E) CRMM2 tumour burden in orthotopic model (*n* ≈ 15). Results are presented as means ± SD from three independent experiments. Representative images show CM tumour burden in the head and tail regions in the ectopic model and a localised tumour in the orthotopic model.

**Figure 8 cancers-12-00587-f008:**
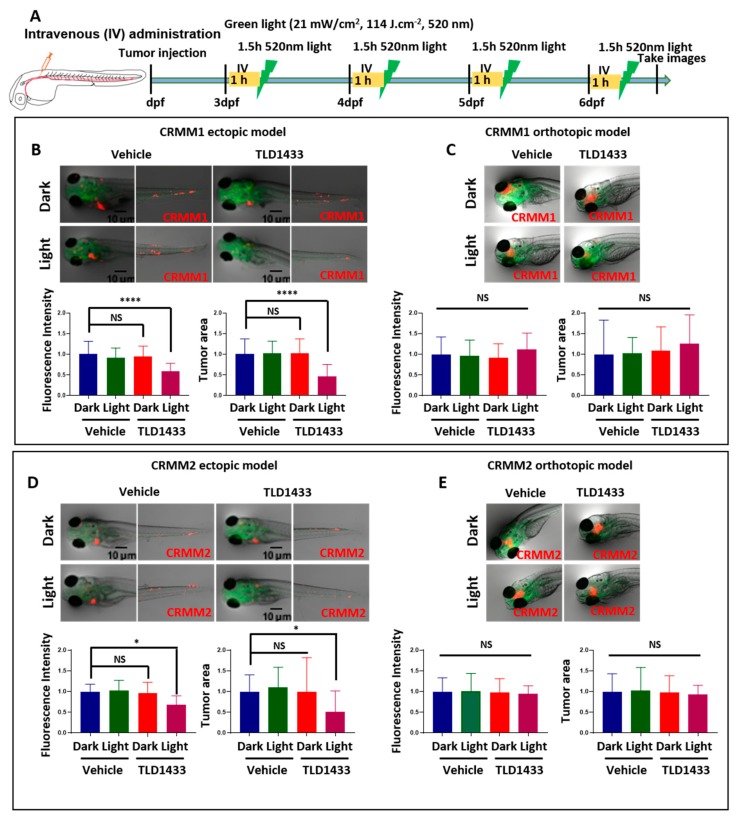
TLD1433 treatment by IV administration in the zebrafish ectopic and orthotopic CM model. (**A**) Schedule of tumour injection and TLD1433 administration in zebrafish embryos. Relative tumour burden was calculated as described in Figure 7. (**B**) CRMM1 tumour burden in the ectopic model (*n* ≈ 30). (**C**) CRMM1 tumour burden in the orthotopic model (*n* ≈ 15). (**D**) CRMM2 tumour burden in the ectopic model (*n* ≈ 30). (**E**) CRMM2 tumour burden in the orthotopic model (*n* ≈ 15). Results are presented as means ± SD from three independent experiments. Representative images show CM tumour burden in the head and tail regions in ectopic model and localised tumours in the orthotopic model.

**Figure 9 cancers-12-00587-f009:**
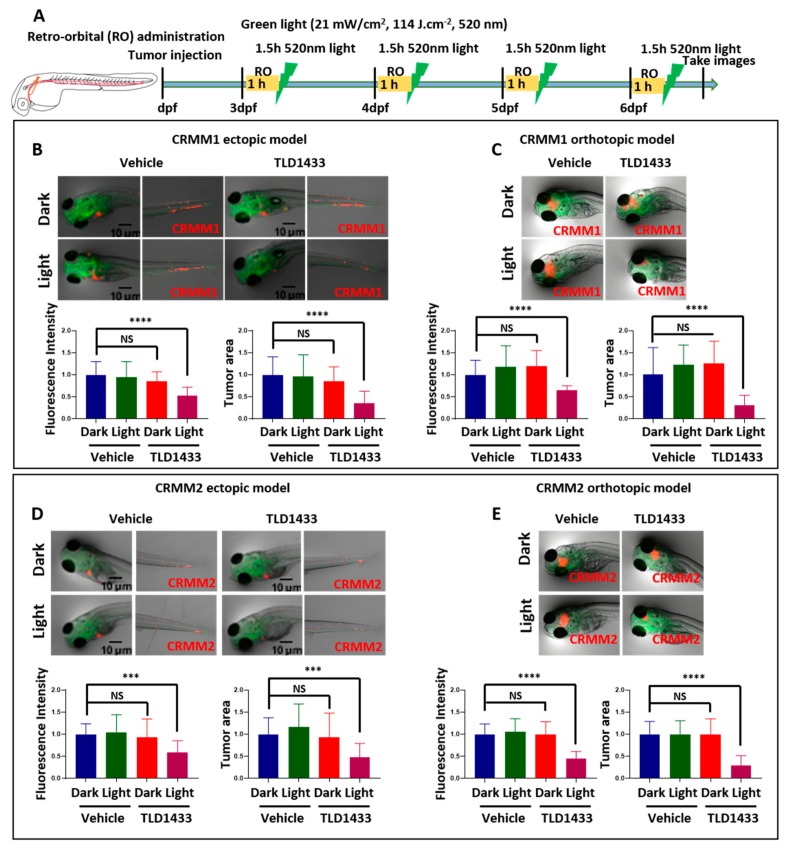
TLD1433 treatment by RO administration in the zebrafish ectopic and orthotopic CM model. (**A**) Schedule of tumour injection and TLD1433 administration in zebrafish embryos. Relative tumour burdens were calculated as described in Figure 7. (**B**) CRMM1 tumour burden in the ectopic model (*n* ≈ 30). (**C**) CRMM1 tumour burden in the orthotopic model (*n* ≈ 15). (**D**) CRMM2 tumour burden in the ectopic model (n ≈ 30). (**E**) CRMM2 tumour burden in the orthotopic model (*n* ≈ 15). Results are presented as means ± SD from three independent experiments. Representative images show CM tumour burden in the head and tail regions in ectopic model and localised tumour in orthotopic model.

**Figure 10 cancers-12-00587-f010:**
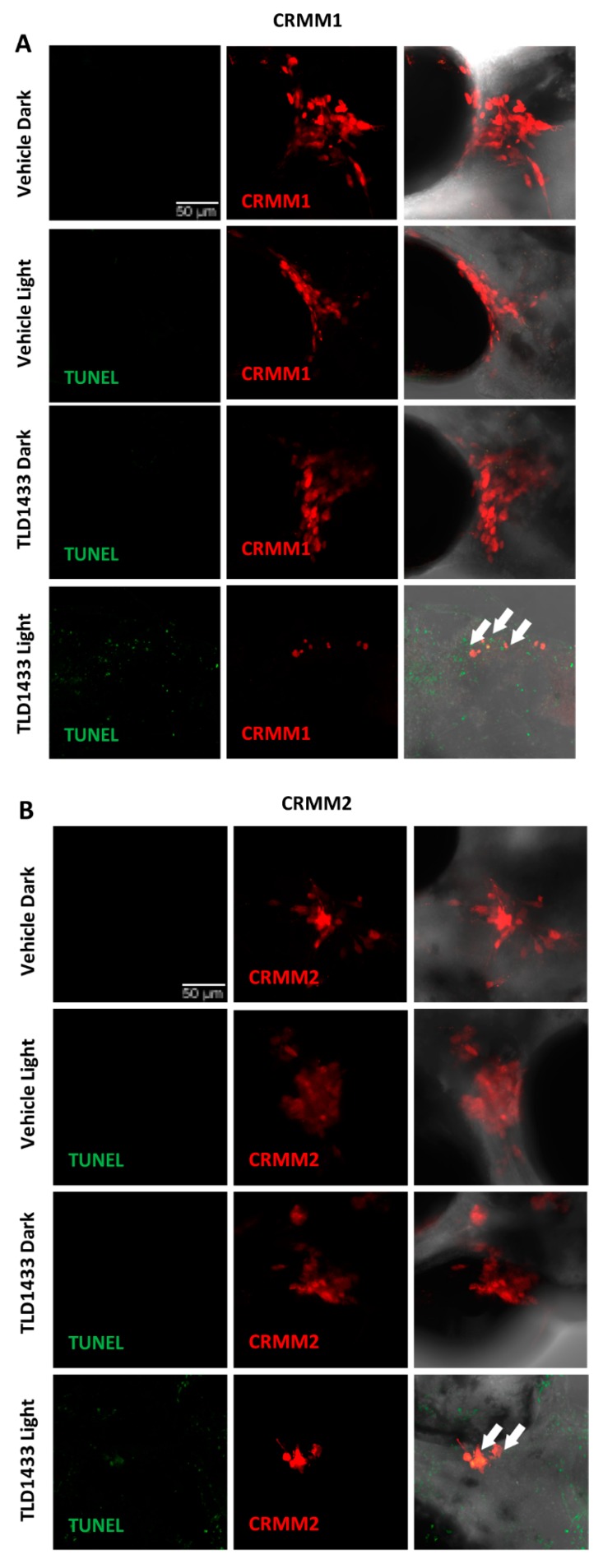
TUNEL assay of in CRMM1 and CRMM2 orthotopic model after RO of TLD1433. Red fluorescent CRMM1 (**A**) and CRMM2 (**B**) cells were injected at 2dpf behind the ZF eye and divided into four groups for drug treatment. RO administration of vehicle control and TLD1433 was performed as described in Figure 9C,E. After dark or light exposure (21 mW/cm^2^, 114 J.cm^−2^, 520 nm) embryos were fixed and TUNEL staining was performed. Representative images of embryos are shown in this figure. (**A**,**B**) In TLD1433 light groups nuclear DNA fragmentation by nucleases is detected by co-localization of green (DNA fragments) and red signal of engrafted CM cells, depicted as yellow signal and marked by white arrows. In control dark, control light, TLD1433 dark, there are no positive green apoptotic tumour cells observed. Background green signal in TLD1433 light groups, does not co-localized with cytosolic red signal, which is diminished in degraded cells and TUNEL stains only the DNA breaks in these CM apoptotic cells. Experiment was performed 3 times with a group size of 10 embryos.

**Figure 11 cancers-12-00587-f011:**
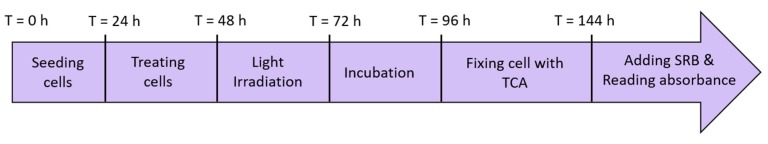
Time line for the SRB assay.

**Table 1 cancers-12-00587-t001:** Three conjunctival melanoma cell lines, three uveal melanoma cell lines and two control cell lines were tested in an SRB cell viability assay, after exposure to TLD1433 with or without 19 J.cm^−2^ green light exposure for 15 min (21 mW/cm^2^, 19 J.cm^−2^). Cell growth inhibition effective concentrations (EC_50_, in µM) and 95% confidence intervals (95% CI, in µM) were obtained. The photo index (PI) was calculated as EC_50_, light divided by EC_50_, dark.

Cell Lines	CRMM1	CRMM2	CM2005.1	OMM1	OMM2.5	MEL270	A431	A375
EC_50_, dark (µM)	0.84	1.0	1.1	1.4	0.64	1.1	>5	>5
95% CI (µM)	−0.23	−0.17	−0.22	−0.48	−0.16	−0.092	n.a	−0.020
	+0.27	+0.19	+0.25	+0.95	+0.19	+0.097	n.a	+0.020
EC_50_,light (µM)	0.0059	0.0048	0.0058	0.014	0.013	0.010	0.049	0.050
95% CI (µM)	−0.00099	−0.00050	−0.00061	−0.0016	−0.0011	−0.0012	−0.025	n.a
	+0.0012	+0.00055	+0.00066	+0.0019	+0.0013	+0.0013	n.a	n.a
PI	140	210	190	100	49	110	>102	>100

**Table 2 cancers-12-00587-t002:** The maximum tolerated dose (MTD) of TLD1433 in wild type zebrafish embryos and in the ectopic and orthotopic CM tumour model.

TLD1433 Administration Type	Maximum Tolerated Dose (MTD)	
	Wild Type Embryos	CM Engrafted Embryos Ectopic and Orthotopic Model
Water	9.2 nM	4.6 nM
Intravenous	4.6 mM	2.3 mM
Retro-orbital	4.6 mM	2.3 mM

**Table 3 cancers-12-00587-t003:** Relative tumour burden quantified by fluorescence intensity in zebrafish embryonic models after treatment with TLD1433, delivered by three different administration routes. The fluorescence intensity is calculated as percentage, compared to the control dark group (100%).

Cell Line	Route of TLD1433 Administration	Relative Tumour Burden as Measured by Fluorescence Intensity					
		Ectopic Model			Orthotopic Model		
		Light Dose (J.cm^−2^)		PI	Light Dose (J.cm^−2^)		PI
		0	114		0	114	
CRMM1	Water	91%	89%	1.0	96%	96%	1.0
	Intravenous	94%	59%	1.6	91%	111%	0.82
	Retro-orbital	85%	53%	1.6	120%	65%	1.8
CRMM2	Water	90%	96%	0.93	96%	104%	0.92
	Intravenous	97%	69%	1.4	98%	95%	1.0
	Retro-orbital	93%	60%	1.6	100%	45%	2.2

**Table 4 cancers-12-00587-t004:** The relative tumour burden quantified by tumour area in zebrafish embryonic models after treatment with TLD1433 delivered by three different administration routes. The tumour area is calculated as percentage, compared to the control dark group (100%).

Cell Line	Route of TLD1433 Administration	Tumour Area					
		Ectopic Model			Orthotopic Model		
		Light dose (J.cm^−2^)		PI	Light dose (J.cm^−2^)		PI
		0	114		0	114	
CRMM1	Water	103%	99%	1.0	90%	89%	1.0
	Intra venous	102%	46%	2.2	109%	126%	0.87
	Retro orbital	85%	36%	2.4	125%	31%	4.1
CRMM2	Water	92%	102%	0.90	104%	95%	1.1
	Intra venous	99%	50%	2.0	97%	93%	1.1
	Retro orbital	94%	48%	2.0	99%	29%	3.4

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
