# Peer review of "TLD1433 Photosensitizer Inhibits Conjunctival Melanoma Cells in Zebrafish Ectopic and Orthotopic Tumour Models"

_cancers, 2020, doi:10.3390/cancers12030587_

Round 1
Reviewer 1 Report
The work: "TLD1433 photosensitizer inhibits conjunctiva melanoma cells in zebrafish ectopic and orthotopic tumour model" is interesting and very well written.
Just some commentaries:
a) I was surprised that authors did not evaluated no biomarker for ROS production;
b) I did not see no ethical concerns nor if the study was approved for the use of animals. This must be included.
Just some minor corrections:
a) In vitro, in vivo, and Danio rerio should be written in italic.
b) The images should be bigger.
Author Response
Dear reviewer,
Thank you very much for your consideration and positive comments concerning our manuscript cancers-721297: TLD1433 photosensitizer inhibits conjunctival melanoma cells in zebrafish ectopic and orthotopic tumour models.
We are pleased to resubmit our revised version. You can find our answers to your questions in the letter below and in the main text of the manuscript depicted by track changes and yellow background.
Major comments:
1) I was surprised that authors did not evaluated no biomarker for ROS production
Answer: Thank you for the valuable comment. TLD1433 is currently under clinical evaluation as PDT photosensitizer in bladder cancer; its singlet oxygen generation quantum yield has been demonstrated to be close to unity, while its ROS generation properties under light irradiation have also been evaluated in different cell lines and animal models. Highlights can be found in the Chem Review article on TLD1433 (reference 19 in our revised manuscript), which we have cited. Since the main focus of our work is the investigation of delivery methods in the zebrafish in vivo model, we did not scrutinize ROS production again.
2) I did not see no ethical concerns nor if the study was approved for the use of animals. This must be included.
Answer: apologies for forgetting this important information. We have provided it in the revised manuscript line: 523-526.
Minor comments:
1) In vitro, in vivo, and Danio rerio should be written in italic.
Answer: We agree, this is corrected.
2) The images should be bigger.
Answer: We agree, the images are enlarged.
By providing all explanations to the valuable comments of reviewers we hope that our revised manuscript will warrant a publication in Cancers.
I look forward to your response,
Yours faithfully,
Prof. Dr. B.E. Snaar-Jagalska
Gorlaeus Laboratory/ Institute of Biology
Einsteinweg 55, 2333 CC, Leiden, The Netherlands
phone: 31-71-5274980
e-mail: B.E.Snaar-Jagalska@biology.leidenuniv.nl

Reviewer 2 Report
I write you in regards to manuscript entitled “TLD1433 photosensitizer inhibits conjunctival melanoma cells in zebrafish ectopic and orthotopic tumour models” which you submitted to Cancers.
As author notes in this report, this study might be useful information for the ruthenium-based photosensitizer TLD1433. This is a well-written and useful contribution, which I think is entirely suitable for publication after revision.
Minor comments
・Page 2, Line 66: 1O2→ 1O2
・Page 2, Line 81: 1O2→ 1O2
・Page 3, Line 128: You should put down with the abbreviation of uveal melanoma.
uveal melanoma cell lines→uveal melanoma (UM) cell lines
・Page 7: In figure 5 B, C, and D, the fonts of vertical and horizontal axes are too small.
Major comments
・Page 14, Line 343~368: As you mentioned in the discussion, the tumour model (ectopic or orthotopic) and the mode of administration of TLD1433 (intravenous or retro-orbital) affect the therapeutic effects of PDT. You should discuss it in greater depth and detail.
・Page 14, Line 369: As you mentioned in the discussion, the zebrafish model is very small. Therefore, tumour tissues are also very small. I think that the tumor size have an effects on the therapeutic effects of PDT. You should also discuss it in greater depth and detail.
Author Response
Dear reviewer,
Thank you very much for your consideration and positive comments concerning our manuscript cancers-721297: TLD1433 photosensitizer inhibits conjunctival melanoma cells in zebrafish ectopic and orthotopic tumour models.
We are pleased to resubmit our revised version. You can find our answers to your questions in the letter below and in the main text of the manuscript depicted by track changes and yellow background.
Major comments:
1) Page 14, Line 343~368: As you mentioned in the discussion, the tumour model (ectopic or orthotopic) and the mode of administration of TLD1433 (intravenous or retro-orbital) affect the therapeutic effects of PDT. You should discuss it in greater depth and detail.
Answer: we agree, but it is unclear to us which specific point the referee wants us to add in the discussion. The whole article discusses how the mode of administration affects the therapeutic effects of PDT. We believe our answer to question 2 of the referee addresses this comment 1) as well.
2) Page 14, Line 369: As you mentioned in the discussion, the zebrafish model is very small. Therefore, tumour tissues are also very small. I think that the tumor size have an effects on the therapeutic effects of PDT. You should also discuss it in greater depth and detail.
Answer: we agree, and as a consequence we have modified our discussion after line 374 on the size of the ZF and its transparency, into:
In ZF embryo models for PDT the small size of the embryo, of the tumour, and the relative optical transparency of all tissues, combine into easy light penetration into the tumour. On the other hand, local irradiation of the tumor is very difficult, and it was not investigated here. Hence, the main challenge in developing a ZF embryo model for testing PDT sensitizers is to ensure that the concentration of the photosensitizer in the tumor tissue is high enough during irradiation yet as low as possible in healthy tissues that will be irradiated as well. This condition is the only way to ensure a maximum dose of oxidative stress in the diseased tissue, while minimizing effects on the rest of the body. In larger animals such as mice or humans, achieving deep light penetration in the tumour tissue and constant light dosimetry can be challenging and requires careful optimization according to the type of cancer, photosensitizer employed, and irradiation wavelength. Nevertheless, the light component is, by definition of PDT, what imparts selectivity for the tumour area, which minimizes global toxicity.
We acknowledge there are advantages and disadvantages to every in vitro and in vivo model for testing anticancer effects, including those from PDT. This is why multiple models are employed for preclinical validation, and the ZF embryo is but one model that will increase our overall understanding of the preclinical efficacy of TLD1433 under different conditions.
Minor comments:
1) Page 2, Line 66: 1O2→ 1O2; Page 2, Line 81: 1O2→ 1O2
Answer: We agree, they are corrected.
2) Page 3, Line 128: You should put down with the abbreviation of uveal melanoma. uveal melanoma cell lines→uveal melanoma (UM) cell lines
Answer: We agree, it is corrected.
3) Page 7: In figure 5 B, C, and D, the fonts of vertical and horizontal axes are too small.
Answer: We agree, the Fonts in figure 5 B, C and D are corrected.
By providing all explanations to the valuable comments of reviewers we hope that our revised manuscript will warrant a publication in Cancers.
I look forward to your response,
Yours faithfully,
Prof. Dr. B.E. Snaar-Jagalska
Gorlaeus Laboratory/ Institute of Biology
Einsteinweg 55, 2333 CC, Leiden, The Netherlands
phone: 31-71-5274980
e-mail: B.E.Snaar-Jagalska@biology.leidenuniv.nl

Reviewer 3 Report
Cancers 2020
“TLD1433 photosensitizer inhibits conjunctival melanoma cells in zebrafish ectopic and orthotopic tumour models”
By Quanchi Chen 1, Vadde Ramu 2, Yasmin Aydar 1, Arwin Groenewoud 1, Xuequan Zhou 2, Martine 5 J. Jager 3, Houston Cole 4, Colin G. Cameron 4, Sherri A. McFarland 4*, Sylvestre Bonnet 2*, B. Ewa 6 Snaar-Jagalska 1
Authors discuss effects of the ruthenium-based photosensitizer (PS) TLD1433 on the treatment of conjunctival melanoma (CM) using a zebrafish cancer model. Authors reported good therapeutic effects when TLD was administrated retro-orbitally (RO). Work is well conducted and only few comments we have: a) it is not completely clear how CM cells treated with TLD die (apoptosis vs necrosis), this is of importance in view of the use of this approach in animal model, when an inflammatory reaction elicited by tumor necrosis may drive several side effects; b) it is also to discuss why IV injection of TLD inhibits only ectopic tumor model, but not orthotopic model, it is very hard to believe that the orthotopically transplanted tumor is not reached by the drug.
Author Response
Dear reviewer,
Thank you very much for your consideration and positive comments concerning our manuscript cancers-721297: TLD1433 photosensitizer inhibits conjunctival melanoma cells in zebrafish ectopic and orthotopic tumour models.
We are pleased to resubmit our revised version. You can find our answers to your questions in the letter below and in the main text of the manuscript depicted by track changes and yellow background.
Major comments:
1) It is not completely clear how CM cells treated with TLD die (apoptosis vs necrosis), this is of importance in view of the use of this approach in animal model, when an inflammatory reaction elicited by tumor necrosis may drive several side effects.
Answer: Thank you very much for your comment and we totally agree with that. 100% discrimination between apoptosis and necrosis is almost impossible. In vitro by Annexin V and Propidium Iodide staining and FACS we clearly showed that in the light-activated TLD1433 group, about half of the cells were found dead, either in the late apoptotic or necrotic quadrant (Figure 2). Importantly, our in situ TUNEL assay on fixed embryos indicated that PDT-driven anti-tumour efficacy of TLD1433 in this PDT regime is partially apoptosis-dependent. This point requires further attention in future research. To address the possible role of the inflammatory reaction elicited by tumor necrosis we modified our discussion after line 385:
However, in this stage of our research we cannot exclude an inflammatory reaction caused exclusively by innate immunity cells (as adaptive immunity is not active yet) of the zebrafish embryos elicited by tumour necrosis or apoptosis after TLD treatment. This reaction may drive same negative side effects or even help to attenuate tumour development in this model [51].
As mentioned in response to Reviewer 2, there are advantages and disadvantages to every in vitro and in vivo model. We would argue that the ZF model would not be the appropriate model to optimize efficacy and the inflammatory response. We would also argue that while Reviewer 3 suggests the inflammatory reaction as a negative consequence, it is required to elicit an adaptive antitumor immune response, which is crucial for improving outcomes for ocular melanoma patients. The benefits of treating any micrometastases through the adaptive immune response far outweigh these potential side effects in a patient population where the disease is almost always fatal. Nevertheless, this would not be the appropriate model for optimizing this immunological benefit.
2) It is also to discuss why IV injection of TLD inhibits only ectopic tumor model, but not orthotopic model, it is very hard to believe that the orthotopically transplanted tumor is not reached by the drug.
Answer: We were also surprised by this, but as we indicated in the text, we observe that IV administration of TLD allows the compound to reach and inhibit CM cells only in the ectopic model. In this model, engrafted cells disseminate through the blood circulation and form small metastatic lesions in the head and tail. We speculate that probably more cells are effectively reached by the drug before activation. In the orthotopic model, lesions are bigger and may be more compact, so that cells are less excisable to the drug. We also cannot exclude involvement of additional biological obstacles, which could lower effective drug concentration reaching CM cells behind the eye.
To clarify the situation, we modified our discussion paragraph as follows after line 405:
Independent of the model, water administration did not to allow for the compound to reach the tumour. For intravenous injection of TLD1433, the situation was model-dependent: the ectopic model showed good activity after light irradiation, while the orthotopic model did not. In the ectopic model, engrafted cells disseminate through the blood circulation and form small metastatic lesions in the head and tail. At this stage, we can only speculate that when TLD1433 is also injected in the blood circulation more cells may effectively be reached by the drug before activation. In the orthotopic model, lesions are bigger and may be more compact, so that cells may be less easily reached by the drug when it is injected in blood. We do know that for other photosensitizers, it can be a challenge to get good distribution in tumours. Notably, local injection of TLD1433 behind the eye resulted in an excellent response for both ectopic and orthotopic models. Overall, the best response was obtained with the orthotopic model combined with local injection of TLD1433, i.e. injection behind the eye. In such a case, maximum drug concentration may be achieved in the tumour right before light irradiation.
In addition to the reviewers comments we also corrected through whole manuscript the light doses (15 minutes green light irradiation (21 mW/cm2, 19 J.cm-2, 520 nm) for cells and 90 minutes green light irradiation (21 mW/cm2, 114 J.cm-2, 520 nm) for zebrafish) used and added the paragraph 4.9. (line 559), which describes the efficacy of TLD1433 by WA, IV and RO in a zebrafish ectopic and orthotopic tumour models.
Fluorescent CRMM1 and CRMM2 cells were injected at 2 dpf into the Duct of Cuvier (ectopic model) and behind the eye (orthotopic model) and TLD1433 was delivered by WA, IV and RO administration with or without a light treatment as described in 4.8. For the WA administration, the 4.6 nM TLD1433 solution was added to the tumour cells injected zebrafish at 2.5, 3.5, 4.5, 5.5 dpf and maintained for 12 h. At 3, 4, 5, 6 dpf, the fish water was refreshed, and after 1 h, embryos were exposed to green light for 90 min (21 mW/cm2, 114 J/cm2, 520 nm). For the IV and RO administration, 1 nL of 2.3 mM TLD1433 solution was injected into via the dorsal vein or the RO site at 3, 4, 5, 6 dpf. After 1 h interval, the embryos were exposed to green light for 90 min (21 mW/cm2, 114 J/cm2, 520 nm). After treatment, the embryos images were required with a Leica M165 FC stereo fluorescence microscope. Tumor growth was quantified by calculating the total fluorescence intensity and area with the ZF4 pixel counting program (Leiden). Each experiment was performed at least 3 times with a group size of >30 embryos.
By providing all explanations to the valuable comments of reviewers we hope that our revised manuscript will warrant a publication in Cancers.
I look forward to your response,
Yours faithfully,
Prof. Dr. B.E. Snaar-Jagalska
Gorlaeus Laboratory/ Institute of Biology
Einsteinweg 55, 2333 CC, Leiden, The Netherlands
phone: 31-71-5274980
e-mail: B.E.Snaar-Jagalska@biology.leidenuniv.nl
